# A Novel Deep Blue LE-Dominated HLCT Excited State Design Strategy and Material for OLED

**DOI:** 10.3390/molecules26154560

**Published:** 2021-07-28

**Authors:** Xuzhou Tian, Jiyao Sheng, Shitong Zhang, Shengbing Xiao, Ying Gao, Haichao Liu, Bing Yang

**Affiliations:** 1State Key Laboratory of Supramolecular Structure and Materials, College of Chemistry, Jilin University, 2699 Qianjin Street, Changchun 130012, China; tianxz19@mails.jlu.edu.cn (X.T.); xiaosb20@jlu.edu.cn (S.X.); yinggao19@mails.jlu.edu.cn (Y.G.); hcliu@jlu.edu.cn (H.L.); 2State Key Laboratory of Inorganic Synthesis and Preparative Chemistry, College of Chemistry, Jilin University, 2699 Qianjin Street, Changchun 130012, China; shengjiyao@jlu.edu.cn

**Keywords:** organic light-emitting diodes, deep blue emitter, LE-dominated HLCT

## Abstract

Deep blue luminescent materials play a crucial role in the organic light-emitting diodes (OLEDs). In this work, a novel deep blue molecule based on hybridized local and charge-transfer (HLCT) excited state was reported with the emission wavelength of 423 nm. The OLED based on this material achieved high maximum external quantum efficiency (EQE) of 4% with good color purity. The results revealed that the locally-excited (LE)-dominated HLCT excited state had obvious advantages in short wavelength and narrow spectrum emission. What is more, the experimental and theoretical combination was used to describe the excited state characteristic and to understand photophysical property.

## 1. Introduction

The organic light-emitting materials play an important role in flat-panel display, solid-state lighting, photodynamic therapy, and so on [1,2,3,4]. These efficient fluorescent molecules often have large π-conjugation plane which could be applied in biological probes and sensors because of good biocompatibility of organic compounds [5,6]. According to the spin-statistics rule, the 75% spin-forbidden triplet excitons will be wasted from the lowest triplet state (T_1_) to the ground state (S_0_) through non-radiative transitions [7,8,9,10]. In order to maximize the utilization of a triplet exciton, a large number of materials were reported including the thermally-activated delayed fluorescence (TADF) materials which were considered to be the most popular of a new generation organic light-emitting diode (OLED) materials [11,12,13]. However, the TADF can promote exciton utilizing efficiency (EUE) by forming a charge-transfer (CT) state, which will cause seriously red-shifted emission at the same time. Further, the separated highest occupied molecular orbital (HOMO) and lowest unoccupied molecular orbital (LUMO) always depends on a twisted donor–acceptor structure or strong CT, leading to broadened electroluminescence (EL) spectra. Lately, Hatakeyama and co-workers reported the B,O-doped polycyclic aromatic molecules with a 1,4-oxaborine substructure identified as multi-resonance TADF (MR-TADF) materials [14,15,16,17,18]. The MR-TADF materials ensured the color purity of emissions under the premise of realizing TADF. However, all of these molecules suffered from an extremely serious loss of raw materials in the synthesis process, causing obstacles to industrial manufacturing. Targeting the above-mentioned problems, the HLCT mechanism had attracted enormous attention because of its advantages [19,20,21]. The HLCT state was hybridized by the LE and CT state when the non-adiabatic LE and CT states had small energy gaps which can cause big mixing coefficients [22]. According to Appendix A, the S_1_ (3.8019 eV) was close to S_2_ (3.9945 eV), indicating a small energy gap in the LE and CT state, which was of benefit to form HLCT. The small energy difference of S_1_-T_5_ was tiny enough (∆E_S1T5_ was calculated to be 0.0768 eV). Based on the tiny ∆E_S1T5_, a “hot exciton” channel can be constructed. With the help of the “hot exciton” channel, T_5_ exciton could be converted to singlet via hRISC with no delayed lifetime of the excited state (less than 10 ns) [23]. Besides, the moderate overlap degree of “hole” and “particle” was different from the typical CT state or LE state. The HLCT materials had unequivocal design principles, aiming to promote the hybridization in different proportions between LE and CT states [24,25,26]. Ma and Hu et al. illustrated that the LE-state was responsible for the high radiative transition rate and high luminous efficiency, while the CT-state was provided with a small splitting and efficient RISC channel, a high PL efficiency and a high EUE at the same time. The HLCT state can achieve high EUE because of the RISC process occurring at high-lying energy levels [27]. As for LE-dominated HLCT, the large weight of the LE state always equals to small reorganization energy. The small reorganization energy represents the suppressed vibrational rotation, resulting in narrow spectrum and deep blue emission. In addition, the HLCT molecules have simple molecular construction, which decreases the difficulty of synthesis and paves the way to mass production.

The 1H-phenanthro[9,10-d]imidazole (PI) is a classic group of HLCT benefiting from its bipolar character which has been systematically investigated by our group in recent years [28,29,30], but the narrow spectrum emission potential of PI is rarely involved. As we all know, the imidazole ring of PI is usually modified in two directions to modulate the new excited state. The substituent can be introduced into the system in a horizontal and vertical direction on the C-position and N-position, respectively. It is also noteworthy that the mild donor and acceptor are important factors to balance the LE and CT components. For example, the triphenylamine (TPA) and cyano group were selected because of the mild electron-donating/withdrawing ability and smaller steric hindrance. In 2015, TBPMCN achieved compatible coexistence between a high photoluminescence quantum yield (PLQY) and high EUE via a quasi-equivalent HLCT state, which was reported by our group [31]. In this work, N-TBPMCN was designed and synthesized, which has a similar structure with TBPMCN. The substituents in C-position and N-position of N-TBPMCN were changed to form LE-dominated HLCT (Scheme 1). As the isomer of TBPMCN, N-TBPMCN demonstrated a deep blue emission and narrowed full width at half maximum (FWHM). This work provided us with a novel molecular design for an LE-dominated HLCT excited state, which was beneficial to realize the electroluminescence with short wavelength and narrow spectrum emission.

## 2. Results and Discussion

### 2.1. Molecular Design

The previous works had shown that the 1H-phenanthro[9,10-d]imidazole (PI) was an excellent candidate to form short wavelength emitting materials [32,33]. Especially for HLCT materials, the PI with electron-withdrawing sp^2^ N-atoms and electron-donating sp^3^ N-atoms was used as the bipolar main body with balanced CT and LE. The cyano group was a common electron acceptor to HLCT. Meanwhile, the cyano group with sp-hybridized N-atoms expanded the conjugation and enhanced the oscillator strength in a vertical direction. Based on these advantages, TBPMCN achieved efficient blue emission successfully because of the HLCT state. In this work, we optimized the combination mode of PI, TPA and cyano group to generate LE-dominated HLCT N-TBPMCN (Scheme 2) in order to realize good color purity and deep blue emission.

The ground state geometry was optimized with Gaussian 09 D. 01 package by the density functional theory (DFT) method at a M062X/6-31g (d, p) level [34]. According to the geometry of the ground state (Figure 1a), the twisting angle of the imidazole ring and the benzene ring in C-position was 25.4°. Because of the small twisting, the C-position can be used to introduce the LE state and increase oscillator strength. As for the N-position, the peripheral hydrogen atoms caused steric hindrance leading to a more twisted angle of 74.7°. Generally speaking, the introduction of the acceptor in the N-position was conducive to induce the CT component, because of the relatively large twisting. However, if we can use the electron donor TPA to replace the acceptor in the N-position, the LE component would be enhanced. In addition, the biphenyl group caused intensive conjugate effects, resulting in a portion of the LE state. The conjugated structure can neutralize the influence of the CT state to some extent.

The natural transition orbital (NTO) was calculated with time-dependent DFT (TDDFT) using a TD-M062X/6-31g (d, p) method, which was used to describe the excited state properties, as shown in Figure 1b. In principle, the common CT state had a totally separated “hole” and “particle”; on the contrary, the LE demonstrated a positive coincident distribution of “hole” and “particle”. On the one hand, the transition from TPA/biphenyl to TPA/biphenyl and the transition from PI to itself can be identified as an obvious LE feature. On the other hand, the benzonitrile played an important role in the D-A structure as electron acceptor, which induced the CT state to some degree. Besides, the transition from TPA/biphenyl to PI can also be observed, representing the CT component of an excited state. In the meantime, the N-TBPMCN had higher S_1_ (3.9775 eV) than TBPMCN (3.8019 eV), and this high value indicated the LE-dominated HLCT state leading to a short wavelength and narrow spectrum emission. The S_1_ oscillator strength of N-TBPMCN was calculated to be 0.7572, which can be expected to possess high PLQY.

### 2.2. Photophysical Characterizations and Excited State Properties

The absorption and emission of N-TBPMCN were investigated in different solvents. The absorption spectra demonstrated b-band absorption of the benzene ring and HLCT absorption at around 254 nm and 351 nm, respectively. The emission presented a fine structure in hexane, and with increasing solvent polarity the fine structure disappeared as shown in Figure 2a and Appendix A. In hexane, the emission wavelength was shorter than 400 nm with the PLQY of 55%. The maximum PLQY was achieved in the ether (77%) because of the well balanced LE and CT of the HLCT state. Further, in acetonitrile solvent, the PLQY dropped down to 1% with the wavelength of 425 nm, indicating that a high polarity environment can damage the hybridization of LE and CT. The solvation effect of N-TBPMCN was obviously distinct from TADF materials, because of its moderate degree of red-shift.

The Lippert–Mataga solvatochromic model revealed the relationship of solvent polarity and Stokes shift, which was used as a common method to estimate the dipole moment of the excited state [35]. The Stokes shift can be calculated by the absorption and emission peaks in different solvents, as shown in Figure 2b, Appendix A. The result of Lippert–Mataga solvatochromic model displayed 14 Debye of the excited state dipole moment, indicating the LE-dominated HLCT state. Based on the smaller dipole moment (17 Debye) of N-TBPMCN than TBPMCN, the solvatochromic model suggested that N-TBPMCN had more LE component than TBPMCN, demonstrating the LE-dominated HLCT state of N-TBPMCN.

As in solution, the non-doped film of N-TBPMCN also emitted within a deep blue region with a wavelength of 426 nm, and the lifetime was measured to be 8.5 ns (Figure 3a,b). As a comparison, we chose N-TBPMCN as guest material and polymethyl methacrylate (PMMA) as host material to prepare doped films. The doped film had a bluer emission with a wavelength of 406 nm and shorter lifetime of 2.3 ns (Figure 3a,b). Besides, the lifetimes of N-TBPMCN in different solutions possessed less than 10 ns. These non-delayed lifetimes indicated an obvious HLCT excited state feature, which can ensure a fast radiative transition rate benefitting the high luminescent efficiency.

### 2.3. Thermal Properties and Electrochemical Properties

The thermal stability of materials is a significant factor to affect electroluminescence (EL) performance. The glass transition temperature (T_g_) and the decomposition temperature of 5 % weight loss (T_d_) were measured to be 218 °C and 475 °C by differential scanning calorimetry (DSC) and thermogravimetric analyses (TGA), respectively. As shown in Figure 4a and Table 1, the results of DSC and TGA demonstrated good thermal stability of N-TBPMCN, which can be evaporated as an emitting layer during device fabrication.

The electrochemical property revealed carrier injection and transport properties, which was important to design the OLED device structure. The highest occupied molecular orbital (HOMO) and lowest unoccupied molecular orbital (LUMO) were calculated by the following equation [36,37]:HOMO (eV) = −[E_ox_ − E(Fc/Fc^+^) + 4.8](1)
LUMO (eV) = −[E_red_ − E(Fc/Fc^−^) + 4.8](2)

According to Figure 4b and Table 1, both HOMO and LUMO were calculated to be −4.86 eV and −2.77 eV. Based on the HOMO and LUMO energy level of N-TBPMCN, the OLED structure can be optimized for the best EL performance.

### 2.4. OLED Performance

Generally, it is necessary to optimize a proper OLED structure for improving device performance [38,39,40,41,42]. All of these OLEDs had a relatively low turn-on voltage of 3.3 V, which is attributed to the suitable device structure with balanced carrier transport despite different doping concentrations. The optimal OLED structure was fabricated as ITO/HATCN (5 nm)/TAPC (25 nm)/TCTA (10 nm)/emitters (20 nm)/TPBI (35 nm)/LiF (8 nm)/Al (500 nm). However, the non-doped OLED of N-TBPMCN suffered from serious aggregation caused by quenching (ACQ) [43,44], leading to a low EQE of 1.4 %. In order to alleviate the ACQ effect, the doped OLED of N-TBPMCN were also fabricated with CBP as host material in the emitter layer. As a result, the host CBP doped with N-TBPMCN (15 wt%) was the most appropriate doping ratio, and this doped OLED harvested the best performance: EQE of 4.0 %, EL wavelength of 423 nm, and FWHM of 50 nm (Table 2). Obviously, the doped OLED demonstrated a shorter emission wavelength, narrower FWHM and higher EQE than those of the non-doped OLED. In Figure 3a and Figure 5a, the PL spectra of N-TBPMCN film had a greater similarity with the EL spectra of doped OLED, rather than non-doped OLED (Appendix A). This indicates that the host material could effectively suppress excimer formation and concentration quenching.

According to previous reports, the deep blue luminescent PPI and mTPA-PPI had similar structures to N-TBPMCN (shown in Appendix A) [26]. Compared to PPI (EQE_max_ = 1.86%) and mTPA-PPI (EQE_max_ = 3.33%), the N-TBPMCN was provided with higher EQE, demonstrating that the benzonitrile played an important role in the D-A system. Although the maximum emission wavelength and FWHM were similar between N-TBPMCN (Figure 5a) and TBPMCN (Appendix A) in non-doped OLEDs, the doped OLED of N-TBPMCN displayed a shorter wavelength and better color purity, compared with those of TBPMCN (EL = 452 nm, FWHM = 93 nm in Appendix A). These results revealed the advantages and potential of the LE-dominated HLCT state in regard to high-efficiency, color-purity and deep blue OLED.

## 3. Materials and Methods

### 3.1. Synthesis of Materials

All of the chemical reagents and solvents were purchased from Acros, Energy Chemical and Changchun Sanbang. The reagents and solvents can be used directly without further purification. The ^1^H NMR spectra and ^13^C NMR spectra of the intermediate products and the target products were illustrated in Appendix A. The synthesis details are as follows:

4-[1-(4-Bromo-phenyl)-1H-phenanthro[9,10-d]imidazol-2-yl]-benzonitrile (N-BrPMCN)

4-Cyanobenzaldehyde (10 mmol, 1.3 g), 9,10-Phenanthrenequinone (10 mmol, 2.1 g), 4-Bromoaniline (40 mmol, 6.9 g) and ammonium acetate (50 mmol, 3.7 g) were added into a 250 mL round-bottom flask. Then, 20 mL CH_3_COOH was chosen as solvent. After three times being degassed, the temperature was set as 120 °C and stirred for 3 h. The solvent was cooled down and the solid particle was separated by vacuum filtration. The filter cake was washed by CH_3_COOH and water in a 2:1 volume ratio to remove soluble impurities. After that, the filter cake was dissolved into CH_2_Cl_2_ and dried by a molecular sieve. The crude product was purified by silica column chromatography eluting by CH_2_Cl_2_ and petroleum ether in a 1:1 volume ratio (R_f_ = 0.3 in CH_2_Cl_2_:petroleum ether = 1:1), and white pure product was obtained (3.1 g, yield = 65 %, 473.05 g/mol). *m*/*z* = 473.58 (M^+^ H^+^)

^1^H NMR (500 MHz, DMSO-d_6_): δ 8.90 (dd, *J* = 30.0 Hz, 5.0 Hz, 2 H, HAr), 8.70 (d, *J* = 5.0 Hz, 1 H, HAr), 7.89 (dd, *J* = 15.0 Hz, 5.0 Hz, 4 H, HAr), 7.80-7.73 (m, 6 H, HAr), 7.61 (t, *J* = 5.0 Hz, 1 H, HAr), 7.44 (t, *J* = 5.0 Hz,1 H, HAr), 7.15 (d, *J* = 5.0 Hz, 1 H, HAr).

4-[1-(4′-Diphenylamino-biphenyl-4-yl)-1H-phenanthro[9,10-d]imidazol-2-yl]-benzonitrile (N-TBPMCN)

N-BrPMCN (5 mmol, 2.4 g), 4-(Diphenylamino)phenylboronic acid (7 mmol, 2.0 g), and K_2_CO_3_ (30 mmol, 4.2 g) were added to a mixed solvent of H_2_O (8 mL), THF (5 mL) and toluene (10 mL) and degassed, Pd(PPh_3_)_4_ (0.22 mmol, 253 mg) was added under a nitrogen atmosphere as a catalytic agent, the temperature was set at 90 °C, and it was stirred for 3 h. Then, the reaction solution was cooled down to room temperature. The solution was extracted with CH_2_Cl_2_ and the organic layer was dried with MgSO_4_. The crude product was purified by silica gel column chromatography eluting by CH_2_Cl_2_ (R_f_ = 0.6 in CH_2_Cl_2_), and yellow pure product was obtained (2.6 g, yield = 78 %, 638.25 g/mol). *m*/*z* = 638.69 (M^+^ H^+^).

^1^H NMR (500 MHz, DMSO-d_6_): δ 8.90 (dd, *J* = 30.0 Hz, 10.0 Hz, 3 H, HAr), 8.70 (t, *J* = 5.0 Hz, 2 H, HAr), 8.01 (d, *J* = 5.0 Hz, 1 H, HAr), 7.94–7.73 (m, 13 H, HAr), 7.60 (d, *J* = 10.0 Hz, 2 H, HAr), 7.47–7.37 (m, 4 H, HAr), 7.26 (d, *J* = 5.0 Hz, 1 H, HAr), 7.11 (m, 4 H, HAr).

^13^C NMR (126 MHz, CDCl_3_) δ 148.32, 147.42, 142.44, 137.83, 137.32, 136.54, 134.69, 134.45, 133.84, 132.31, 132.10, 132.00, 130.50, 129.73, 129.55, 129.45, 129.15, 128.91, 128.81, 128.56, 128.50, 128.17, 127.84, 127.59, 127.54, 126.91, 126.85, 126.65, 126.56, 126.22, 126.11, 125.65, 125.52, 125.35, 124.79, 124.48, 124.36, 124.24, 124.05, 123.57, 123.47, 123.22, 122.98, 122.74, 122.55, 122.44, 122.38, 121.07, 120.98, 120.78, 120.58, 118.52, 118.40, 112.41, 112.18.

Anal. calculated for C_65_H_42_N_6_: C, 86.49; H, 4.73; N, 8.77; found: C, 86.38; H, 4.73; N, 8.77.

### 3.2. Photophysical Measurements

The solutions were diluted to 1 × 10^−5^ mol·L^−1^ to ensure the monodispersion of the molecular. All the solutions were put into the quartz cell and solid samples were put into a quartz plate in order to ensure accuracy. The UV-3100 spectrophotometer was applied to record the UV-vis absorption spectra. The fluorescence measurements were carried out with an RF-5301PC. The PLQY of doped and non-doped films were measured by using an Edinburgh FLS-980 with integrating sphere apparatus. An Edinburgh FLS-980 with an EPL-375 optical laser was the main instrument to estimate lifetime. The samples were put into the quartz plate to estimate lifetime. The total lifetimes of multi-sectioned PL-decay spectra were calculated using the following equation:
(3)$$\tau =\frac{\sum_{i=1}^{n}\tau_{i}^{2}A_{i}}{\sum_{i=1}^{n}\tau_{i}A_{i}}$$
where τ is lifetime; *i* represents for the number of the lifetime components; and *A_i_* is the proportion of each lifetime components.

### 3.3. Electrochemical Measurement

The cyclic voltammetry (CV) was measured by a BAS 100W Bioanalytical System. The three electrode system was introduced to conduct the electrochemical measurement. The glass carbon disk (Φ = 3 mm), platinum wire and Ag/Ag^+^ electrode were used as working electrode, auxiliary electrode and reference electrode, respectively. The ferrocenium/ferrocene was set as a redox couple. The solution was blown up with nitrogen for 5 min to exclude oxygen for more accurate results.

### 3.4. OLED Fabrication and Performances

The substrate of OLED was set as ITO-coated glass with a sheet resistance of 20 Ω square^−1^. We chose deionized water, isopropyl alcohol, acetone and chloroform to wash the ITO glass by ultrasonic cleaner. The OLED was fabricated by evaporation. The organic layers were controlled at the rate of 0.03–0.1 nm/s, the LiF layer was controlled at the rate of 0.01 nm/s and the Al layer was controlled at the rate of 0.3 nm/s. The PR650 spectra scan spectrometer was used to record EL spectrum and 2T model 2400 programmable voltage-current source.

The HATCN is 2,3,6,7,10,11-Hexacyano-1,4,5,8,9,12-hexaazatriphenylene. The TAPC is 4,4′-cyclohexylidenebis[N,N-bis(p-tolyl)aniline]. The TCTA is 4,4′,4″-Tris(carbazol-9-yl)-triphenylamine. The TPBI is 1,3,5-Tris(1-phenyl-1H-benzo[d]imidazol-2-yl)benzene. The host material CBP is 1,4-di(9H-carbazol-9-yl)benzene.

## 4. Conclusions

In summary, a new structure N-TBPMCN was designed and synthesized for a deep blue OLED emitter. The LE-dominated HLCT excited state property was revealed by photophysical characterizations and quantum chemical calculation. Compared to the isomer TBPMCN, the weight of the LE component of N-TBPMCN was enhanced by exchanging the position of TPA and cyano groups. The OLED of N-TBPMCN harvested shorter wavelength emission and narrower FWHM, because of the advantages of LE-dominated HLCT. The N-TBPMCN was provided with good thermal properties and electrochemical properties, which was appropriate for electroluminescence OLED. Overall, we reported a novel material with deep blue emission and high color purity by simple synthetic route in this work, which contributes to the molecular design of better color purity blue-emissive materials.

## Data Availability

The data presented in this study are available on request from the corresponding author.

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
