# Peer review of "A Novel Deep Blue LE-Dominated HLCT Excited State Design Strategy and Material for OLED"

_molecules, 2021, doi:10.3390/molecules26154560_

Round 1
Reviewer 1 Report
The manuscript of Tian et al. describes the synthesis and photophysical characterisation of a new emitter molecule. The practical work has been competently carried out and the material is sufficiently characterised. The manuscript is well structured and the formal presentation is of high standard. Although there are many language issues, the manuscript is easy to read. However, there is some room for improvement:
#A thorough language polishing is necessary.
#Introduction: The HLCT concept should be better elaborated. Only from the description given in this section, it does not become clear how this concept works and how the triplet excitons formed in a device can be utilized by HLCT emitter molecules. An energy level diagram illustrating the involved process would greatly enhance the understanding of the matter.
#Scheme 1: In the diagram or in the caption it should be specified if the luminescence spectra were measured in solution (solvent?) or as solid.
#I would recommend shifting scheme 2 to chapter 2.1. and briefly explain the synthesis of the molecule.
#Calculations: Again, an energy level diagram would help a lot to understand the electronic structure of the molecule. Which states are LE, which are CT in nature, which of them have singlet or triplet character and finally which are those LE and CT states which mix with each other allowing for ISC/rISC? As the authors present their molecule as an optimised version of a recent reported one, they should also include the data of the original molecule for comparison. By comparing these two molecules in a clearly arranged illustration, they could support they statement that they really have changed the balance between LE and CT in the new molecule.
#Photophysical characterisation: Did the authors measure also emission quantum yields in solution? These values would be certainly interesting.
I am not sure if the authors use the term "doped" in the correctly. They report on doped films and do not specify the host material. It reads like they would dope N-TBPMCN with the host, but it should be vice versa (see p. 4, line 158). In the figure caption of Fig. 3 they should also specify the host.
A similar issue can be found in the section on OLED fabrication. Also here, the host should be specified and it should be written, that “… the host XYZ is doped with N-TBPMCN (XY %) …”
#Figure 4: The solvent which was used for the CV measurement should be given. The signal representing the ferrocenium/ferrocene redox couple should also be marked.
#page 5, line 185: It should read “ferrocenium”. Can the authors assign the oxidation/reduction to a part of the molecule (imidazole, biphenyl, diphenylamine?)? Why is the reduction irreversible? Is the oxidation reversible (peak separation?)? The description of the experiment and instrumentation should be shifted to section 3.
#page 5, lines 189/190: Please give the literature for the calculation of the HOMO/LUMO level, particularly the value 4.8 eV has to be substantiated by a reference.
In summary, the manuscript describes interesting results and thus could be published after major revision indicated above.
Author Response
Response to Reviewer 1 Comments
Point 1: A thorough language polishing is necessary.
Response 1: Thanks for the reviewer’s advice. The manuscript has been polished.
Point 2: Introduction: The HLCT concept should be better elaborated. Only from the description given in this section, it does not become clear how this concept works and how the triplet excitons formed in a device can be utilized by HLCT emitter molecules. An energy level diagram illustrating the involved process would greatly enhance the understanding of the matter.
Response 2: Thanks for the reviewer’s advice. The concept of HLCT has been added into the introduction “The HLCT state was hybridized by LE and CT state when the non-adiabatic LE and CT states had small energy gap which can cause big mixing coefficient. According to Figure S1 and Table S1, the S1 (3.8019 eV) was close to S2 (3.9945 eV) indicating small energy gap of LE and CT state which was of benefit to form HLCT. The small energy difference of S1-T5 was tiny enough (∆ES1T5 was calculated to be 0.0768 eV). Based on the tiny ∆ES1T5, “hot exciton” channel can be constructed. With the help of “hot exciton” channel, T5 exciton could be converted to singlet via hRISC with no delayed lifetime of the excited state (less than 10 ns). Besides the moderate overlap degree of “hole” and “particle” was different from typical CT state or LE state. The HLCT materials had unequivocal design principle, aiming to promote the hybridization in different proportions between LE and CT states. Hu and Ma et al. illustrated that the LE-state was responsible for the high radiative transition rate and high luminous efficiency while the CT-state was provided with small splitting and efficient RISC channel, a high PL efficiency and a high EUE in the same time. The HLCT state can achieve high EUE because of the RISC process occurring at high-lying energy levels.”
And the energy level diagram and chart have been applied in the paper.
Figure S1 The energy landscape for excited states of N-TBPMCN and TBPMCN.
Table S1 The specific energy value of singlet and triplet for N-TBPMCN and TBPMCN calculated by density functional theory (DFT) method at a M062X/6-31g (d, p).
N-TBPMCN (eV) |
TBPMCN (eV) |
||||||
S1 |
3.9775 |
T1 |
3.1218 |
S1 |
3.8019 |
T1 |
3.1218 |
S2 |
4.0195 |
T2 |
3.2242 |
S2 |
3.9945 |
T2 |
3.3271 |
S3 |
4.3269 |
T3 |
3.5819 |
S3 |
4.2326 |
T3 |
3.6657 |
S4 |
4.3803 |
T4 |
3.7009 |
S4 |
4.3805 |
T4 |
3.7283 |
S5 |
4.569 |
T5 |
3.9007 |
S5 |
4.4296 |
T5 |
3.7723 |
S6 |
4.6401 |
T6 |
3.9313 |
S6 |
4.5396 |
T6 |
3.8656 |
S7 |
4.669 |
T7 |
3.9666 |
S7 |
4.558 |
T7 |
3.9284 |
S8 |
4.7193 |
T8 |
4.0494 |
S8 |
4.6657 |
T8 |
3.9825 |
S9 |
4.8828 |
T9 |
4.221 |
S9 |
4.801 |
T9 |
4.1304 |
S10 |
4.9666 |
T10 |
4.3136 |
S10 |
4.8451 |
T10 |
4.2123 |
Point 3 Scheme 1: In the diagram or in the caption it should be specified if the luminescence spectra were measured in solution (solvent?) or as solid.:
Response 3: Thanks for the reviewer’s advice. The caption has been corrected as “Scheme 1 Molecular design of blue-shifted, narrower-band-emissive N-TBPMCN and the EL spectra of N-TBPMCN and TBPMCN in solid.”
Point 4 I would recommend shifting scheme 2 to chapter 2.1. and briefly explain the synthesis of the molecule.
Response 4: Thanks for the reviewer’s advice. The scheme 2 has been shifted to chapter 2.1. And we have added explain for the synthesis as “Scheme 2 The synthesis route of N-TBPMCN. The intermediate N-BrPMCN was synthetized by Debus–Radziszewski reaction. And by Suzuki reaction the N-BrPMCN was used to synthetize the N-TBPMCN.”
Point 5 Calculations: Again, an energy level diagram would help a lot to understand the electronic structure of the molecule. Which states are LE, which are CT in nature, which of them have singlet or triplet character and finally which are those LE and CT states which mix with each other allowing for ISC/rISC?
Response 5: Thanks for the reviewer’s advice. The energy level diagram and table have been added in the above part.
In addition, the NTOs were also provided in the following part. According to the NTOs, S1 of N-TBPMCN had more CT component, and the S2 had more LE component. The hybridization of LE state and CT state generated S1 and S2. The energy gap between T5 and S1 is tiny enough (0.0768 eV). The T5 exciton is possible transformed into singlet exciton through the “hot exciton” channel via hRISC.
Point 6 As the authors present their molecule as an optimised version of a recent reported one, they should also include the data of the original molecule for comparison. By comparing these two molecules in a clearly arranged illustration, they could support they statement that they really have changed the balance between LE and CT in the new molecule.
Response 6: Thanks for the reviewer’s advice. The TBPMCN was calculated by the same calculation conditions (TD-M062X/6-31g (d, p)). And we have given a more detailed explanation in the article as following “the N-TBPMCN had higher S1 (3.9775 eV) than TBPMCN (3.8019 eV), and this high value indicated the LE-dominated HLCT state leading to short wavelength and narrow spectrum emission. The S1 oscillator strength of N-TBPMCN was calculated to be 0.7572 which can be expected to possess high PLQY.”
Point 7 Photophysical characterisation: Did the authors measure also emission quantum yields in solution? These values would be certainly interesting.
Response 7: Thanks for the reviewer’s advice.
The quantum yields have been added including the measurement details as following;
Table S3 PLQY of N-TBPMCN in different solvents.
Solvents |
Hexane |
Ether |
THF |
Acetonitrile |
PLQY (%) |
55 |
73 |
14 |
1 |
3.3 Relative PLQY in solution
The PLQY in solution is with 0.1 mol/L quinine sulfate aqueous solution for reference. The PLQY of quinine sulfate aqueous solution is 0.546, and the PLQY of sample can be calculated by following equation:
the QYs is relative quantum yield, the Is and Iq are the area of sample and quinine sulfate aqueous solution emission spectrum, the As and Aq are the absorbance of sample and quinine sulfate aqueous solution at 365 nm.
We explained the mechanism of PLQY changing in the paper: “In hexane, the emission wavelength was shorter than 400 nm with the PLQY of 55%. The maximum PLQY was achieved in the ether (77%) because of the well balanced LE and CT of the HLCT state. And in acetonitrile solvent, the PLQY dropped down to 1% with the wavelength of 425 nm, indicating that high polarity environment can damage the hybridization of LE and CT.”
Point 8 I am not sure if the authors use the term "doped" in the correctly. They report on doped films and do not specify the host material. It reads like they would dope N-TBPMCN with the host, but it should be vice versa (see p. 4, line 158). In the figure caption of Fig. 3 they should also specify the host.
Response 8: Thanks for the reviewer’s advice. We have corrected these problems.
“As a comparison, we chose N-TBPMCN as guest material and polymethyl methacrylate (PMMA) as host material to prepare doped film. The doped film had bluer emission with the wavelength of 406 nm and shorter lifetime of 2.3 ns.”
“Figure 3 (a) The PL spectra of N-TBPMCN in doped and non-doped (PMMA as host material) films; (b) The lifetimes of N-TBPMCN in doped and non-doped films.”
Point 9 A similar issue can be found in the section on OLED fabrication. Also here, the host should be specified and it should be written, that “… the host XYZ is doped with N-TBPMCN (XY %) …”
Response 9: Thanks for the reviewer’s advice. We noted the host material “In order to alleviate the ACQ effect, the doped OLED of N-TBPMCN were also fabricated with CBP as host material in the emitter layer. As a result, the host CBP doped with N-TBPMCN (15 wt%) was the most appropriate doping ratio, and this doped OLED harvested the best performance: EQE of 4.0 %, EL wavelength of 423 nm, and FWHM of 50 nm.”
Point 10 Figure 4: The solvent which was used for the CV measurement should be given. The signal representing the ferrocenium/ferrocene redox couple should also be marked.
Response 10: Thanks for the reviewer’s advice. The solvents for CV were measured using Anhydrous Tetrabutylammonium Hexafluorophosphate as electrolyte in CH2Cl2 (positive scan) and N,N-Dimethylformamide (negative scan).
We have remade the electrochemical measurement of the N-TBPMCN as following
Figure 4 (a) The TGA and DSC of N-TBPMCN; (b) The cyclic voltammetry (CV) curve of N-TBPMCN.
In addition the ferrocenium/ferrocene redox couple is measured again, and demonstrated as following.
Based on the ferrocenium/ferrocene redox couple, we have updated the value of HOMO and LUMO in Table 1.
Table 1 The thermal and electrochemical properties of N-TBPMCN materials.
Compounds |
Td/Tg (oC) |
HOMO (eV) |
LUMO (eV) |
Energy gap (eV) |
N-TBPMCN |
475/218 |
-4.97 |
-2.74 |
2.23 |
Point 11 page 5, line 185: It should read “ferrocenium”. Can the authors assign the oxidation/reduction to a part of the molecule (imidazole, biphenyl, diphenylamine?)? Why is the reduction irreversible? Is the oxidation reversible (peak separation?)? The description of the experiment and instrumentation should be shifted to section 3.
Response 11: Thanks for the reviewer’s advice. We have corrected the wrong spelling of ferrocenium.
The imidazole, biphenyl, diphenylamine have similar peaks of the oxidation/reduction. It is difficult to assign the oxidation/reduction to a part of the molecule.
The peaks of the oxidation/reduction are reversible and the first scan and the second scan doesn’t show severe decay. And the image is demonstrated as following.
The description of the experiment and instrumentation has been shifted to section 3 as following.
3.3 Electrochemical Measurement
The cyclic voltammetry (CV) was measured by BAS 100W Bioanalytical System. The three electrode system was introduced to conduct the electrochemical measurement. The glass carbon disk (Φ = 3 mm), platinum wire and Ag/Ag+ electrode were used as working electrode, auxiliary electrode and reference electrode respectively. The ferrocenium /ferrocene was set as redox couple. The solution was blown up with nitrogen for 5 min to exclude oxygen for more accurate results.
Point 12 page 5, lines 189/190: Please give the literature for the calculation of the HOMO/LUMO level, particularly the value 4.8 eV has to be substantiated by a reference.
Response 12: Thanks for the reviewer’s advice. These articles can be used as references.
Tang X, Liu H, Liu F, et al. Efficient Red Electroluminescence From Phenanthro [9, 10‐d] imidazole‐Naphtho [2, 3‐c][1, 2, 5] thiadiazole Donor‐Acceptor Derivatives[J]. Chemistry–An Asian Journal, 2021.
Liu F, Liu H, Tang X, et al. Novel blue fluorescent materials for high-performance nondoped blue OLEDs and hybrid pure white OLEDs with ultrahigh color rendering index[J]. Nano Energy, 2020, 68: 104325.

Reviewer 2 Report
The authors present an OLED based on a new material. I think that the work is suitable for publication. Despite that I think that some aspects must be improved. The presentation of the works reported in the literature should be improved to better show the degree of novelty of the work. The authors did not compare the value that obtain for the EQE with the values reported in the literature.
The indication of the Material and Methods must be improved:
What was the excitation wavelength used to measure the PLQY?
How the lifetime values where estimated? I suppose that was performed a fit. More details should be added.
Other minor errors:
In line 61 “PLQY” is used without being defined.
In line 70 it is written “ermissive”. It should be replaced by "emissive".
In line 105 the authors claim, “which can be expected to possess high PLQY.” Why should this happen?
Table S1 includes abbreviations that were not defined.
In Figure S4, the label of the peaks is to far away from the peaks not allowing a good identification of the peaks.
Author Response
Response to Reviewer 2 Comments
Point 1: #The authors present an OLED based on a new material. I think that the work is suitable for publication. Despite that I think that some aspects must be improved. The presentation of the works reported in the literature should be improved to better show the degree of novelty of the work. The authors did not compare the value that obtain for the EQE with the values reported in the literature.
Response 1: Thanks for the reviewer’s advice.
We have added the comparison of the EQE of these molecules as following:
“According to previous report, the deep blue luminescent PPI and mTPA-PPI had similar structure with N-TBPMCN (shown in Figure S3) [26]. Compared to PPI (EQEmax = 1.86%) and mTPA-PPI (EQEmax = 3.33%), the N-TBPMCN was provided with higher EQE demonstrating the benzonitrile played an important role in the D-A system.”
The structure of PPI and mTPA-PPI were added into the supporting information.
Point 2: The indication of the Material and Methods must be improved:
What was the excitation wavelength used to measure the PLQY?
Response 2: Thanks for the reviewer’s advice. The excitation wavelength was set as 347 nm same as maximum absorption wavelength of the molecule.
Point 3: How the lifetime values where estimated? I suppose that was performed a fit. More details should be added.
Response 3: Thanks for the reviewer’s advice. The details of the photophysical measurement including the lifetime measurement were added as following “An Edinburgh FLS-980 with an EPL-375 optical laser was the main instrument to estimate lifetime. The total lifetimes of multi-sectioned PL-decay spectra are calculated using following equation:
where τ is lifetime; i represents for the number of the lifetime components; and Ai is the proportion of for each lifetime components.”
Point 4: In line 61 “PLQY” is used without being defined.
Response 4: Thanks for the reviewer’s advice. The PLQY has been defined as “photoluminescence quantum yield (PLQY)”
Point 5: In line 70 it is written “ermissive”. It should be replaced by "emissive".
Response 5: Thanks for the reviewer’s advice. The word has been corrected as emissive.
Point 6: In line 105 the authors claim, “which can be expected to possess high PLQY.” Why should this happen?
Response 6: Thanks for the reviewer’s advice. The oscillator strength is positively correlated to PLQY the relative high oscillator strength can cause high PLQY.
Point 7: Table S1 includes abbreviations that were not defined.
Response 7: Thanks for the reviewer’s advice. The abbreviations have been defined.
Point 8: In Figure S4, the label of the peaks is to far away from the peaks not allowing a good identification of the peaks.
Response 8: Thanks for the reviewer’s advice. The new figures have been corrected.
